# Tropospheric Volcanic SO$_2$ Mass and Flux Retrievals from Satellite. The Etna December 2018 Eruption

Stefano Corradini [1,*], Lorenzo Guerrieri [1], Hugues Brenot [2], Lieven Clarisse [3], Luca Merucci [1], Federica Pardini [4], Alfred J. Prata [5,6], Vincent J. Realmuto [7], Dario Stelitano [1] and Nicolas Theys [3]

1    Istituto Nazionale di Geofisica e Vulcanologia (INGV), ONT, 00143 Rome, Italy; lorenzo.guerrieri@ingv.it (L.G.); luca.merucci@ingv.it (L.M.); dario.stelitano@ingv.it (D.S.)
2    Royal Belgian Institute for Space Aeronomy (BIRA-IASB), 1180 Uccle Brussels, Belgium; hugues.brenot@oma.be
3    Spectroscopie de l'Atmosphère, Service de Chimie Quantique et Photophysique, Université Libre de Bruxelles, 1050 Brussels, Belgium; lclariss@ulb.ac.be (L.C.); nicolast@aeronomie.be (N.T.)
4    Istituto Nazionale di Geofisica e Vulcanologia (INGV), Sezione di Pisa, 56125 Pisa, Italy; federica.pardini@ingv.it
5    AIRES Pty. Ltd., Mt Eliza 3930, Australia; fred@aires.space
6    Visiting Professor, School of Electrical Engineering, Computing and Mathematical Sciences, Curtin University, Perth 6845, Australia
7    Jet Propulsion Laboratory, California Institute of Technology, Mail Stop 183-501, 4800 Oak Grove Drive, Pasadena, CA 91109, USA; vincent.j.realmuto@jpl.nasa.gov
*    Correspondence: stefano.corradini@ingv.it; Tel.: +39-06-51860621

**Abstract:** The presence of volcanic clouds in the atmosphere affects air quality, the environment, climate, human health and aviation safety. The importance of the detection and retrieval of volcanic SO$_2$ lies with risk mitigation as well as with the possibility of providing insights into the mechanisms that cause eruptions. Due to their intrinsic characteristics, satellite measurements have become an essential tool for volcanic monitoring. In recent years, several sensors, with different spectral, spatial and temporal resolutions, have been launched into orbit, significantly increasing the effectiveness of the estimation of the various parameters related to the state of volcanic activity. In this work, the SO$_2$ total masses and fluxes were obtained from several satellite sounders—the geostationary (GEO) MSG-SEVIRI and the polar (LEO) Aqua/Terra-MODIS, NPP/NOAA20-VIIRS, Sentinel5p-TROPOMI, MetopA/MetopB-IASI and Aqua-AIRS—and compared to one another. As a test case, the Christmas 2018 Etna eruption was considered. The characteristics of the eruption (tropospheric with low ash content), the large amount of (simultaneously) available data and the different instrument types and SO$_2$ columnar abundance retrieval strategies make this cross-comparison particularly relevant. Results show the higher sensitivity of TROPOMI and IASI and a general good agreement between the SO$_2$ total masses and fluxes obtained from all the satellite instruments. The differences found are either related to inherent instrumental sensitivity or the assumed and/or calculated SO$_2$ cloud height considered as input for the satellite retrievals. Results indicate also that, despite their low revisit time, the LEO sensors are able to provide information on SO$_2$ flux over large time intervals. Finally, a complete error assessment on SO$_2$ flux retrievals using SEVIRI data was realized by considering uncertainties in wind speed and SO$_2$ abundance.

**Keywords:** satellite remote sensing; volcanic monitoring; SO$_2$ mass and flux retrievals; Etna eruption

## 1. Introduction

During their degassing and eruptive activities, volcanoes emit large quantities of gases and particles into the atmosphere. Among the different released gases, H$_2$O, CO$_2$ and SO$_2$ are the most abundant [1,2]. In particular, the SO$_2$ emitted into the atmosphere affects air quality [3–6], the environment [7,8], climate [9–12] and human health [13–17]. Volcanic SO$_2$ can be also used as a proxy for volcanic ash [18,19], which is extremely dangerous

for aircraft engines [13]. The $SO_2$ flux yields insights into the magmatic processes that control volcanic activity during both quiescent and eruptive phases [20,21] and gives information on magma-gas separation depths, conduit structure and magma pressure [2,22]. By knowing the gas mass ratio (e.g., $CO_2/SO_2$), $SO_2$ flux can also be used to estimate the fluxes of other volcanic species, thus improving our understanding of the global volcanic volatile budgets [23–25]. Moreover, the temporal variation of $SO_2$ flux can be used as a precursor of volcanic eruptions [21,26–28]. For all these reasons, there is great interest in improving the quality and frequency of volcanic $SO_2$ mass and flux measurements in real time.

In the last two decades, the technological advances from satellite remote sensing systems have marked a major step forward in the monitoring of volcanic eruptions. Satellite-based remote sensing offers a unique way by which volcanic emissions can be monitored on a global scale and gives the only opportunity to monitor the emissions of volcanoes in the most remote regions of the earth. Nowadays, a variety of geostationary/polar (GEO/LEO), multispectral/hyperspectral (multi/hyper) satellite sensors are used for the detection and retrieval of volcanic $SO_2$ by exploiting its ultraviolet (UV) and thermal infrared (TIR) absorption features, such as the Total Ozone Mapping Spectrometer (TOMS; LEO, hyper, UV) [29–31], the Global Ozone Monitoring Experiment-2 (GOME/GOME-2; LEO, hyper, UV) [32–34], the SCanning Imaging Absorption spectroMeter for Atmospheric CartograpHY (SCIAMACHY; LEO, hyper, UV) [35], the Ozone Monitoring Instrument (OMI; LEO, hyper, UV) [36–38], the Hyperspectral Infrared Atmospheric Sounder (IASI, LEO, hyper, TIR) [39–42], the Atmospheric Infrared Sounder (AIRS; LEO, hyper, TIR) [43], the Advanced Spaceborne Thermal Emission and Reflection Radiometer (ASTER; LEO, multi, TIR) [44–47], the Moderate Resolution Imaging Spectroradiometer (MODIS; LEO, multi, TIR) [48–51], the Visible Infrared Imaging Radiometer Suite (VIIRS; LEO, multi, TIR) [52], the Spinning Enhanced Visible and InfraRed Imager (SEVIRI; GEO, multi, TIR) [49,51,53,54] and HIMAWARI (GEO, multi, TIR) [55,56]. Recently, the Tropospheric Monitoring Instrument (TROPOMI; LEO, hyper, UV) [57,58] was launched onboard the Sentinel-5 Precursor satellite, and with its high sensitivity (and pixel resolution compared to the other hyperspectral sensors), it increases significantly the possibility of revealing previously undetectable $SO_2$.

In addition to the different retrieval strategies developed for volcanic $SO_2$ columnar abundance estimation, several approaches have also been introduced for $SO_2$ flux estimation. The "traverse" approach is based on the definition of transects perpendicular to the plume axis and considering constant or variable wind speeds [46,49,51,59]; the "delta-M" method [60,61] is based on the mass conservation equation applied to different $SO_2$ masses obtained by successive satellite overpasses; the "inversion" method uses atmospheric models that also provide information on pixel altitudes [62,63]. All procedures have advantages and drawbacks (for an overview, see [64]): the traverse method is the simplest and gives reliable results when the wind speed is known and its variation in space is not significant; the delta-M method requires a continuous emission in time or large plumes; the inversion approach can be considered more accurate, but it is highly dependent on the considered meteorological dataset [63].

In the past years, only a handful of cross-comparison exercises have been realized by considering volcanic $SO_2$ total masses and fluxes retrieved from satellite systems [18,49,50,64] and from satellites and ground-based systems [46,51,58,59,63].

In this work, volcanic $SO_2$ total masses and flux retrievals from the six satellite instruments, namely the GEO SEVIRI and the LEO MODIS, VIIRS, TROPOMI, IASI and AIRS, are compared for a single eruption event for the first time. The $SO_2$ columnar abundances are computed by applying different procedures in different spectral ranges, while the fluxes are obtained by means of the "traverse" approach. In the case of near simultaneous LEO measurements of the same instrument, the $SO_2$ flux is obtained by integrating the different contributions. Finally, a complete $SO_2$ flux error assessment is

realized for SEVIRI, considering the uncertainties of wind speed and $SO_2$ abundance. As a test case, the tropospheric and low-ash-content Christmas 2018 Etna eruption is considered.

Etna is a multi-crater stratovolcano located on the east coast of Sicily in southern Italy (see Figure 1). It is one of the most active volcanoes in the world and one of the strongest sources of $SO_2$ during and between eruptions [65].

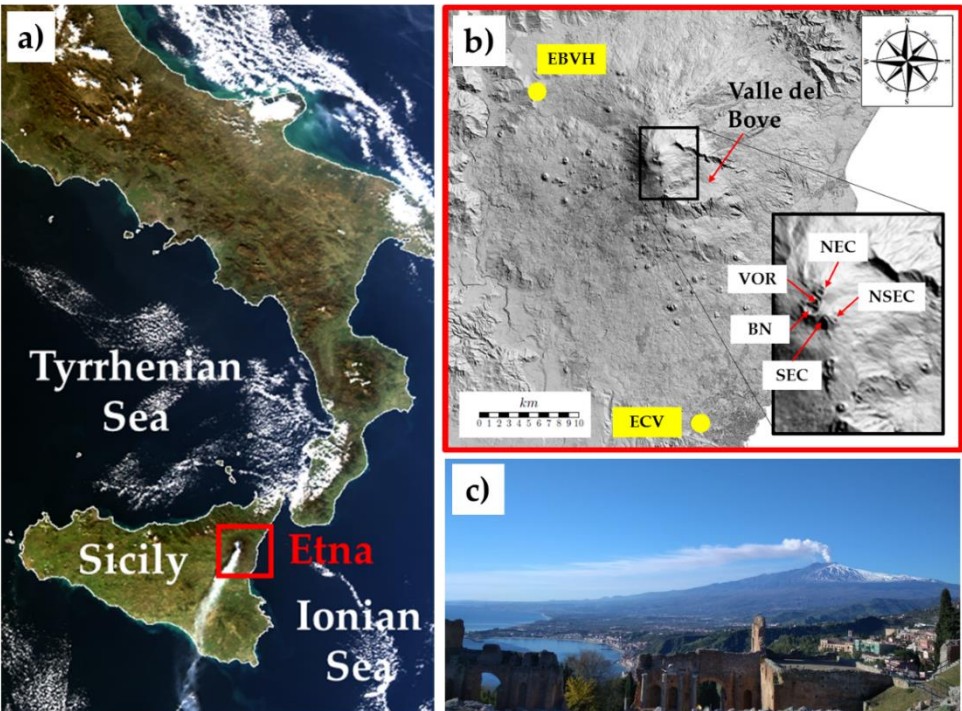

**Figure 1.** (**a**) Image of southern Italy collected from the MODIS satellite sensor on board the Terra/Aqua NASA satellite the 27 December 2018 at 12:20 UTC. (**b**) Map of Etna volcano with indication of the ground-based visible cameras (yellow points). In the zoom, the main active craters are indicated: Voragine (VOR), Bocca Nuova (BN), North East Crater (NEC), South East Crater (SEC) and New SEC (NSEC). (**c**) Etna activities of 27 December 2018 as seen from Taormina. Images modified from Corradini et al., 2020.

An intense sequence of explosive activities, from strombolian to lava fountains, occurred with significant frequency starting in 2011. On the morning of 24 December 2018, the moderate explosive activity and small lava flows at the summit craters already present [66–68] suddenly increased, producing a long eruptive fracture on the southeastern flank of the volcano [69]. Among the lava flow, at about 11:00 UTC, a strong ash/gas column was produced and ejected up to about 8 km above the sea level (asl). Following the wind field, the volcanic cloud was dispersed south-eastward, causing disruption at Catania International Airport. The major explosive activity decreased in the afternoon of 24 December, but consistent ash/$SO_2$ emissions continued until 30 December [51].

The article is organized as follows: in Section 2, the satellite systems and the $SO_2$ detection and retrieval procedures applied to the different instrument measurements are summarized. The cross-comparison strategy is presented in Section 3, and the results are presented in Section 4. In Section 5, a complete $SO_2$ flux error assessment, considering SEVIRI data, is realized, while in Section 6 a discussion of the results is presented. Final conclusions are drawn in Section 7.

## 2. Satellite Instrument Characteristics, $SO_2$ Mass and Flux Retrieval Procedure Description

In this work, different satellite instruments, such as SEVIRI, MODIS, VIIRS, TROPOMI, IASI, and AIRS were considered. These instruments, on geostationary and polar satellite

platforms, from multispectral to hyperspectral, have different temporal and spatial resolutions and exploit the $SO_2$ absorption at different spectral ranges. Table 1 summarizes the main satellite instruments and volcanic $SO_2$ algorithm characteristics considered in this work, and the pixel sizes for the different satellite instruments, considering their nadir view, are represented in Figure 2. The figure emphasizes that the ground pixel differences are significant, ranging from fine spatial resolutions associated with multispectral instruments (VIIRS, MODIS and SEVIRI) to coarse spatial resolution for the hyperspectral instruments (TROPOMI, IASI and AIRS).

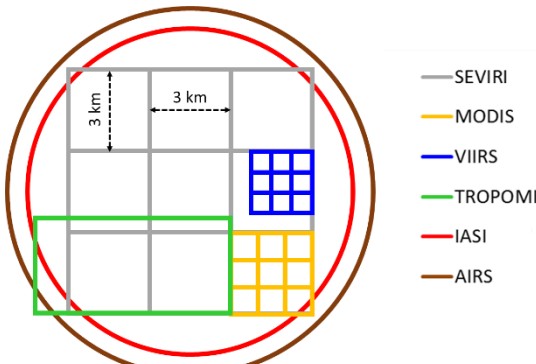

**Figure 2.** Nadir view ground pixel sizes for the different satellite instruments considered in the cross-comparison.

As Table 1 shows, despite the lower spatial resolution, the hyperspectral instruments have the highest sensitivities. Particularly significant is TROPOMI, able to detect $SO_2$ columnar abundance of about 0.02 g/m$^2$ (0.7 DU), i.e., about thirty times better than the sensitivity of the multispectral satellite sensors. The spectral absorption signature considered for the $SO_2$ retrievals varies from UV (TROPOMI) to TIR at 7.3 μm (IASI and AIRS) and 8.7 μm (SEVIRI, MODIS and VIIRS). Among the different sensitivities, the different spectral ranges imply the presence of atmospheric species that could affect the retrievals. As an example, the $SO_2$ retrieval in the UV is affected by the $O_3$ absorption, while in the TIR the 7.3 μm is highly affected by atmospheric water vapor and the 8.7 μm $SO_2$ absorption lies inside a wide atmospheric window (spectral region with high transmittance). The presence of volcanic ash may affect all the measurements (for the multispectral instruments, refer to [49,70]; for hyperspectral instruments, refer to [40,71–74]). As the table shows, the retrieval techniques adopted for the $SO_2$ estimation are significantly different, from simplified approaches based on the computation of atmospheric transmittances (Volcanic Plume Retrieval—VPR) [75–77] to the use of a cache atmospheric spectra for re-use within a scene (Plume Tracker—PT) [52] to Differential Optical Absorption Spectroscopy (DOAS) [57,58], Spectral Matching and Optimal Estimation [78]. All the described differences in satellite sensors and retrieval algorithm characteristics make the cross-comparison extremely significant. The retrieval detection limits are derived from [58] and slightly modified for the multispectral instruments in agreement with [46,49,50].

**Table 1.** Satellite sensors and SO$_2$ retrieval algorithm characteristics. SO$_2$ detection limits adapted from [58]. [(*)] 0.1 DU is for the UTLS and 5 DU is for the lower troposphere/boundary layer. [(**)] V. Realmuto, personal communication.

| Sensor | Satellite | Orbit | Type | Spectral Range | Spatial Resolution (km$^2$) | Temporal Resolution (@ Midlatitudes) (Per Day) | Algorithm for SO$_2$ retrievals (*) = Operational Products | SO$_2$ Absorption Band | Detection Limit (t/km$^2$) (DU) | Errors (%) |
|---|---|---|---|---|---|---|---|---|---|---|
| SEVIRI | MSG (EUMETSAT) | GEO | Multi | VIS-TIR (0.6–13.4 µm,12 bands) | 3 × 3 | 96 or 288 (15 or 5 min) | VPR (Volcanic Plume Retrieval) [75–77] | TIR (8.7 µm) | 0.5 (20) | 40 [50,79] |
| MODIS | Terra/Aqua (NASA) | LEO | Multi | VIS-TIR (0.6–14.4 µm, 36 bands) | 1 × 1 (TIR) | 2–4 | VPR (Volcanic Plume Retrieval) [75–77] | TIR(8.7 µm) | 0.5 (20) | 40 [50,79] |
| VIIRS | NPP/NOAA20 (NASA) | LEO | Multi | VIS-TIR (0.6–12 µm, 22 bands) | 0.75 x 0.75 (TIR) | 2–4 | PT (Plume Tracker) [52] | TIR (8.7 µm) | 0.5 (20) | 15 [(**)] |
| TROPOMI | Sentinel5p (ESA) | LEO | Hyper | UV-SWIR (270–500 nm; 675–775 nm; 2305–2385 nm) | 3.5 × 7.2 (3.5 × 5.5 since 6 August 2019) | 1–2 | DOAS (Differential Optical Absorption Spectroscopy) [57,58] | UV (312–326 nm, 325–335 nm, 360–390 nm) | 0.009–0.02 (0.3–0.7) | 35 [58] |
| IASI | Metop-A/B (EUMETSAT) | LEO | Hyper | IR (3.6–15.5 µm) | Circular, 12 km diameter | 4 | LUT (look-up-table) [39,40] | TIR (7.3 µm) | 0.003–0.1 [(*)] (0.1–5) | 50 [40] |
| AIRS | Aqua (NASA) | LEO | Hyper | IR (3.7–6.6 µm; 8.8–15.4 µm; 6.2–8.2 µm) | Circular, 13.5 km diameter | 1–2 | Spectral Matching and Optimal Estimation [78] | TIR (7.3 µm) | 0.2 (6) | ±6 DU (rms) [78] |

The uncertainty estimation of $SO_2$ abundance retrieval, obtained by applying the VPR approach to SEVIRI and MODIS measurements, has not yet been carried out in detail. For this reason, the retrieval error inserted in Table 1 takes into account the uncertainty of the $SO_2$ retrieval, using similar procedures based on look-up tables—LUT [50] and the intrinsic uncertainty induced by the application of a simplified linearization model as in VPR itself [79], giving a total uncertainty of 40%. Considering the high $SO_2$ and low ash content that characterized this event, the TROPOMI $SO_2$ retrieval uncertainty is set to 35%, while for IASI an error of 50% is fixed. For IASI, as the retrieval exploits the 7.3 μm $SO_2$ band, the detection limit varies greatly with the altitude of the $SO_2$ plume. In the lower troposphere (0–4 km), it ranges from undetectable to 1 DU, depending on the amount of water vapor and thermal contrast [80]; however, with increasing altitude, the detection limit drops and can be as low as 0.1 DU in the upper troposphere/lower stratosphere [41]. The VIIRS and AIRS $SO_2$ retrieval errors were set to 15% and $\pm$ 6 DU (Realmuto private communication and [78]). Among the different sources of uncertainty, the Volcanic Plume Top Height (VPTH) is what induces the highest errors in the $SO_2$ retrievals. In this work, the VPTHs considered are those obtained from the ground-based VIS camera network [81] installed around Etna (see Figure 1 in this paper and Table 1 in [51]): 4.0 km the 26th, 4.5 km the 27th, 29th and 30th and 5.5 km the 28th. These VPTH values were used for all the satellite retrievals, except for IASI, which uses its own estimations.

The $SO_2$ flux is obtained from all the satellite $SO_2$ retrieval images by considering the traverse method and then applying the following equation:

$$F(t) = l * v(t) * \sum_{i=1}^{n} M_i \qquad (1)$$

where l is the transect width (m), v is the wind speed (m/s) and $M_i$ is the $SO_2$ columnar abundance (kg/m$^2$) for a certain pixel of the transect.

As described in [51], the $SO_2$ flux from SEVIRI data is computed considering the transect placed at a distance of 30 $\pm$ 1.5 km from the summit craters. This distance guarantees the minimization of the retrieval uncertainties induced by both the opacity of the pixels close to the craters and the dilution of the pixels far from the craters. The flux is derived by processing one SEVIRI image every 15 min, thus obtaining the time series for the entire eruptive period. The main drawback of such a procedure is that for wind speed greater than 3.3 m/s (3 km/15 min), the use of a single transect could cause the loss of some flux features (due to volcanic puffs, for example). The wind speed is computed from the interpolation between the VPTH values used for the $SO_2$ retrievals and the daily mean wind speed profiles. The latter are obtained as the mean of the wind speed profiles collected in a 6 h step (00, 06, 12, 18 UTC) and derived from the ARPA (Agenzia Regionale per la Protezione Ambientale) database [82]. Knowing the wind speed, the $SO_2$ flux at 30 km is then reported to 0 km (over the vents), the reference start for all the fluxes.

For the LEO satellite systems, the $SO_2$ flux is computed by applying the traverse method, and the wind speed is the same daily average considered for SEVIRI. Different from SEVIRI in this case, all the transects perpendicular to the volcanic cloud axis are considered [46,49,51,59,64]. Depending on wind speed and plume extension, this will make it possible to obtain an $SO_2$ flux trend for several hours before the image acquisition. When acquisitions of the same satellite sensor appear in rapid succession (see Figure 3 for MODIS, VIIRS, TROPOMI and IASI), the flux is computed as the mean of the fluxes obtained from the single images at the same time.

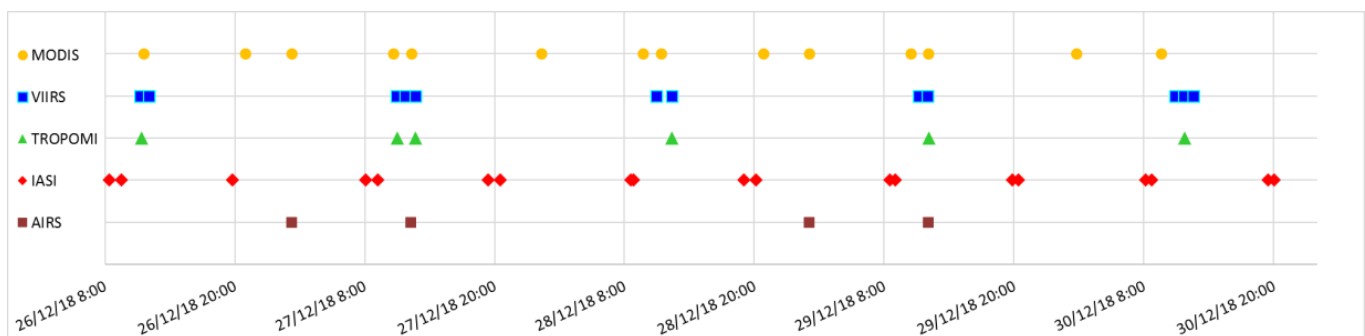

**Figure 3.** Time distribution of the LEO satellite data processed.

## 3. Data Available and Cross-Comparison Strategy Description

Table 2 summarizes the satellite data available for the cross-comparison. In Figure 3, the same images are displayed in a time scale (here, only the LEO platforms are shown, knowing the GEO-SEVIRI data were processed continuously during the whole Etna Christmas 2018 eruption, every 15 min). As Table 2 and Figure 3 show, Terra/Aqua day/night MODIS data are considered, while the images of VIIRS on board NPP/NOAA-20 are processed during the daytime. One TROPOMI measurement is available every day, except 27 December (two images), while four images per day are available for IASI (Metop-A/Metop-B). Finally, four images are considered for AIRS.

**Table 2.** List of GEO and LEO satellite data processed for the cross-comparison exercise. Note that TROPOMI and IASI are both operational products.

| Instruments | Images Processed |
|---|---|
| SEVIRI (400 images) | From 26 December @ 08:00 UTC to 30 December @ 12:00 UTC, every 15 min |
| MODIS (14 images) | Aqua, 26/12 11:35 UTC; Terra, 26/12 21:00 UTC; Aqua, 27/12 01:15 UTC; Terra, 27/12 10:40 UTC; Aqua, 27/12 12:20 UTC; Aqua, 28/12 00:20 UTC; Aqua, 28/12 11:25 UTC; Terra, 28/12 09:45 UTC; Terra, 28/12 20:50 UTC; Aqua, 29/12 01:05 UTC; Terra, 29/12 10:30 UTC; Aqua, 29/12 12:05 UTC; Aqua, 30/12 01:45 UTC; Terra, 30/12 09:35 UTC |
| VIIRS (12 images) | NPP, 26/12 11:18 UTC; N20, 26/12 12:06 UTC; NPP, 27/12 11:00 UTC; N20, 27/12 11:48 UTC; NPP, 27/12 12:42 UTC; N20, 28/12 11:00 UTC; NPP, 28/12 12:18 UTC; N20; 09/12 11:12 UTC; NPP, 29/12 12:00 UTC; N20, 30/12 10:54 UTC; NPP, 30/12 11:42 UTC; N20, 30/12 12:36 UTC |
| TROPOMI (6 images) | 26/12 11:23 UTC; 27/12 11:03 UTC; 27/12 12:43 UTC; 28/12 12:23 UTC; 29/12 12:08 UTC; 30/12 11:48 UTC (from NRT product) |
| IASI (19 images) | IASI-A, 26/12 08:27 UTC; IASI-B, 26/12 09:33 UTC; IASI-A, 26/12 19:47 UTC; IASI-A, 27/12 08:06 UTC; IASI-B, 27/12 09:12 UTC; IASI-A, 27/12 19:26 UTC; IASI-B, 27/12 20:32 UTC; IASI-A, 28/12 08:36 UTC; IASI-B, 28/12 08:52 UTC; IASI-A, 28/12 19:05 UTC; IASI-B, 28/12 20:11 UTC; IASI-B, 29/12 08:31 UTC; IASI-A, 29/12 09:04 UTC; IASI-B, 29/12 19:50 UTC; IASI-A, 29/12 20:24 UTC; IASI-B, 30/12 08:10 UTC; IASI-A, 30/12 08:44 UTC; IASI-B, 30/12 19:30 UTC; IASI-A, 30/12 20:04 UTC (from NRT product) |
| AIRS (4 images) | 27/12 01:18 UTC; 27/12 12:18 UTC; 29/12 01:06 UTC; 29/12 12:06 UTC |

Table 2 and Figure 3 indicate that many images are coincident. Note that, instead of the whole 2018 Christmas Etna eruption (24–30 December), only the period between 26–30 December is considered in this work. The reasons for this choice are twofold: only a few images were collected on 24, and between 25 and 26 December at about 08:00 UTC, with a wide meteorological system affecting the Etnean area, making the retrievals extremely difficult, if not impossible [51].

Since the sensors sensitivity is significantly different, it is necessary to define an area where the volcanic clouds are detected from all the systems. This area is identified as 34–38N and 14–18E (see red rectangle in Figure 4). This is the region where SEVIRI (the

sensor with the lower sensitivity per unit of area) identified all the volcanic clouds in the period considered for the cross-comparison (26–30 December). All the pixels contained in the defined area are considered for the $SO_2$ mass and flux computations.

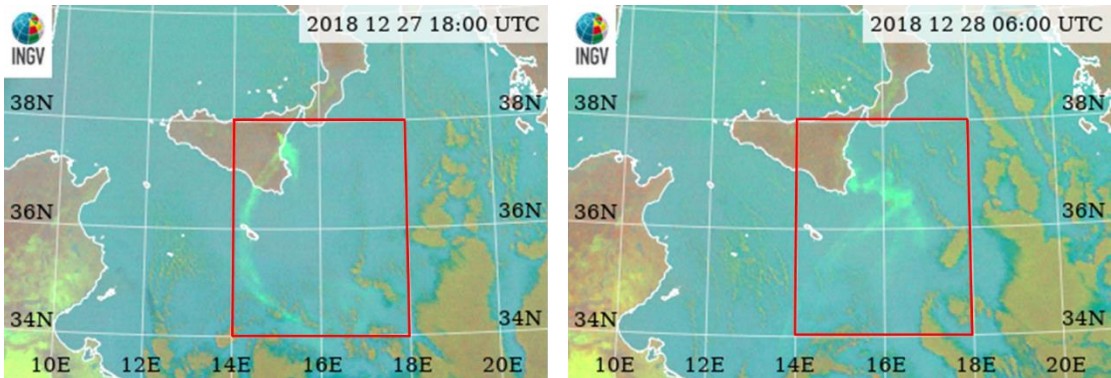

**Figure 4.** Red rectangle: area selected for the $SO_2$ total mass and flux cross-comparison. The two satellite images collected on 27 and 28 December 2018 at 18:00 and 06:00 UTC, respectively, are representative of the higher volcanic cloud extension detected from SEVIRI in the whole 26–30 December period.

To simplify and standardize the calculation of the different quantities considered in the cross-comparison, the LEO satellite retrieval images were all georeferenced in the same GEO SEVIRI latitude–longitude grid, previously resampled to $3 \times 3$ km$^2$.

## 4. Results

In this section, the volcanic cloud $SO_2$ areas and total masses and fluxes are presented for all the GEO and LEO satellite sensors.

### 4.1. SO_2 Total Mass and Area

Figure 5 shows an example of the $SO_2$ columnar abundance products from the different instruments, for near simultaneous satellite images acquired on 27 December from 12:18 to 12:45 UTC (except IASI, which was at 09:12 UTC). The figure emphasizes the ability of all the different systems to detect and retrieve volcanic $SO_2$ and their significantly different sensitivity: higher for TROPOMI, IASI and AIRS and lower for SEVIRI, MODIS and VIIRS. This difference is emphasized in particular in the distal part of the cloud, where the hyperspectral sensors are able to detect low $SO_2$ amounts where multispectral cannot. Almost all the pixels detected from the multispectral sensors have $SO_2$ columnar abundance greater than 0.5 g/m$^2$, while about 95% of pixels detected from the hyperspectral sensors have $SO_2$ abundance lower than this threshold. The maximum $SO_2$ amount retrieved from VIIRS turns out to be significantly higher than the maximum obtained from the other multispectral sensors (SEVIRI/MODIS) in the region close to the vents. A possible reason is that the PT algorithms used for the VIIRS processing do not correct for the effect of volcanic ash, in contrast to the VPR procedure used for the SEVIRI/MODIS processing.

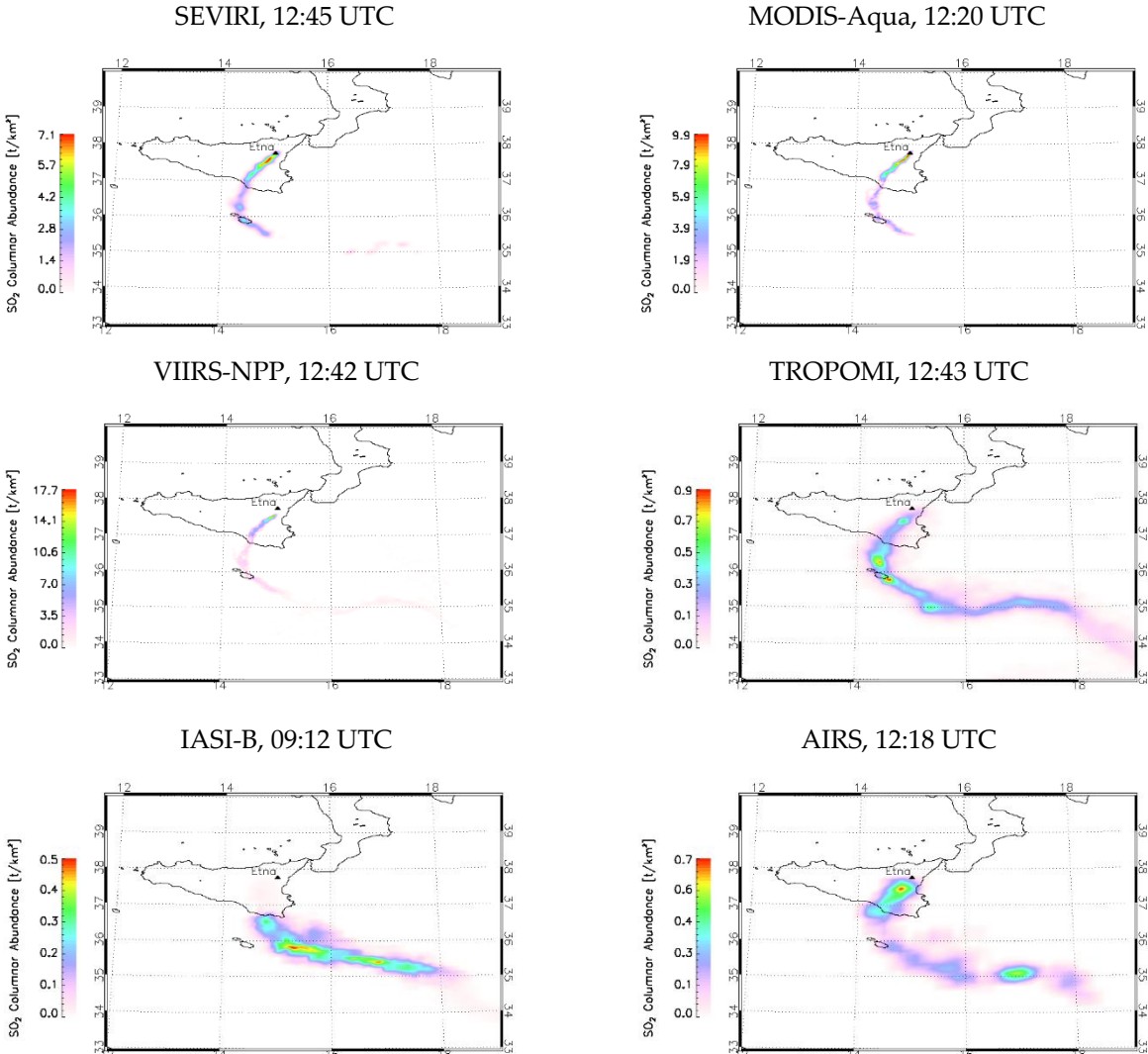

**Figure 5.** Near simultaneous $SO_2$ maps obtained from the different satellite measurements, collected on 27 December 2018 at around 12:30 UTC (except IASI, collected at 09:12 UTC).

Figures 6 and 7 show the time series of the volcanic cloud $SO_2$ areas and total amounts of time series, respectively, obtained from the different satellite sensors. To be sure to consider only the pixel part of the volcanic cloud, the volcanic cloud areas are computed considering the pixels with $SO_2$ abundance greater than 1 DU. Figure 6 confirms the much higher sensitivity for TROPOMI and IASI, which ranges from two to four times the areas detected from the multispectral sensors.

The $SO_2$ total mass uncertainties shown in Figure 7 are those summarized in Table 1. As the figure shows, the $SO_2$ total masses are in good agreement, indicating that the low $SO_2$ abundances per pixel, retrieved from hyperspectral sensors (see Figure 5), are compensated by the greater volcanic cloud area detected (see Figures 5 and 6). The total $SO_2$ masses lie within the retrieval errors for all the different instruments, except for IASI on 28 December PM, for which the $SO_2$ total amounts are significantly higher than the values obtained from SEVIRI. After 27 December PM, the comparison shows that from 28 to 30 December, the IASI $SO_2$ total masses are higher than those obtained from SEVIRI.

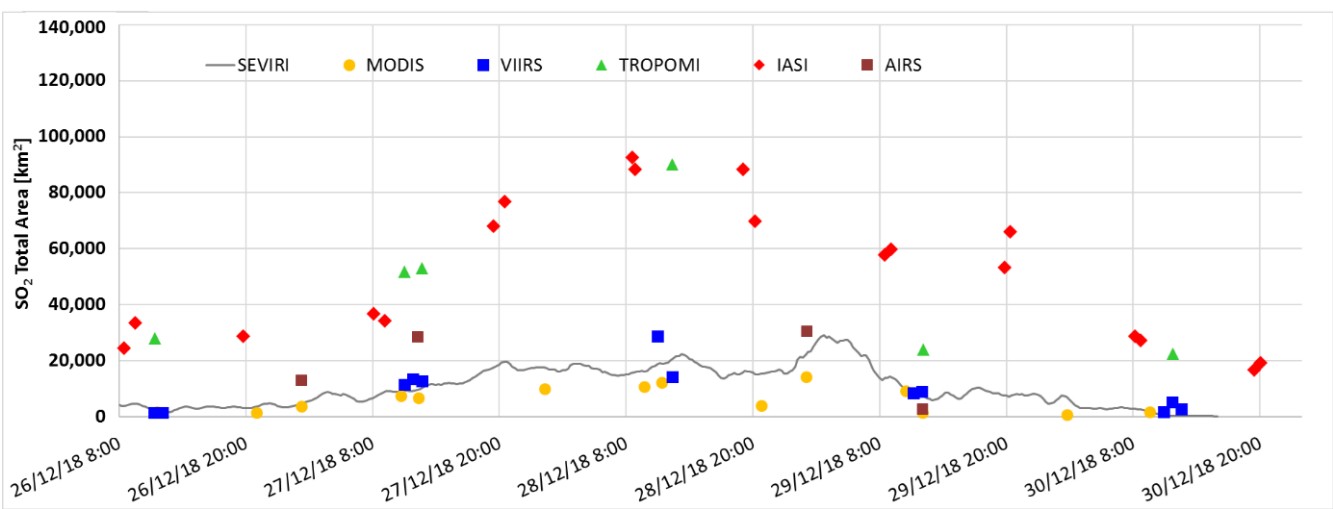

**Figure 6.** SO$_2$ total area computed from the images collected by the different satellite instruments in the latitude–longitude grid 34–38N, 14–18E.

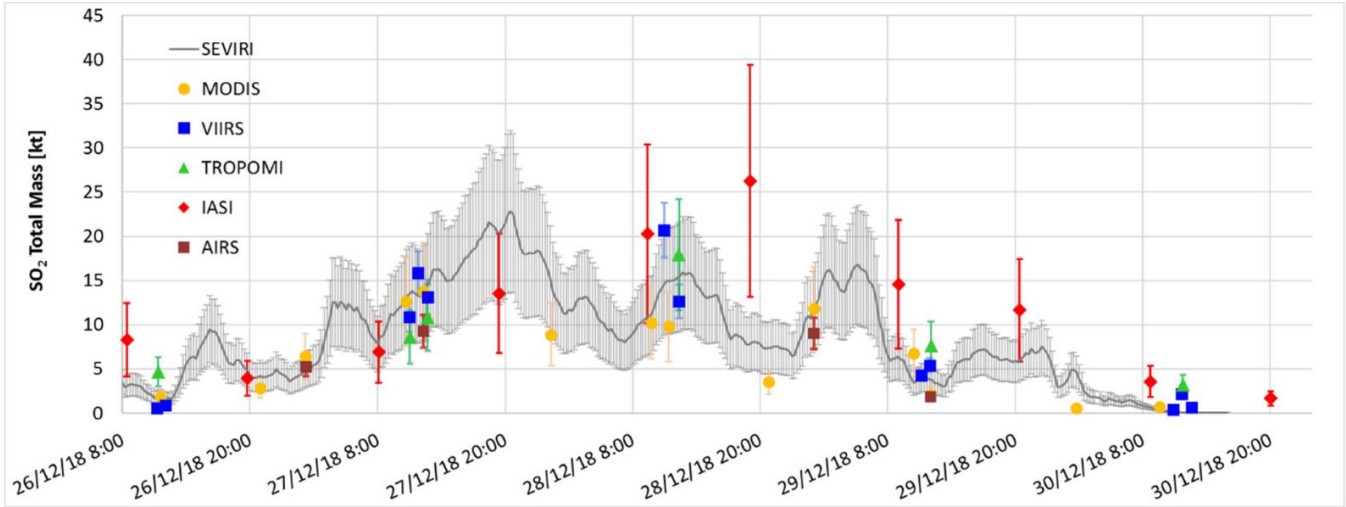

**Figure 7.** SO$_2$ total masses computed from the images collected by the different satellite instruments in the latitude–longitude grid 34–38N, 14–18E.

As mentioned in Section 3, an important source of error in SO$_2$ retrieval is the uncertainty on VPTH. Figure 8 shows the plume altitudes used as input for the SEVIRI, MODIS, VIIRS and TROPOMI SO$_2$ processing (gray line), and for IASI (red rhombus). The latter are the mean values (with standard deviations) obtained from the IASI retrievals, while the altitudes used for all the other instruments are those obtained from the ground-based VIS camera network installed on Etna. For the camera retrievals, a nominal uncertainty of ± 500 m is considered [81]. Both VPTH estimations are affected by uncertainties: the ground-based cameras estimations are realized in the visible spectral range; the nighttime VPTH are simply the daily values extended. On the other hand, from 26 to 30 December, the volcanic cloud was significantly diluted, thus affecting the satellite retrievals.

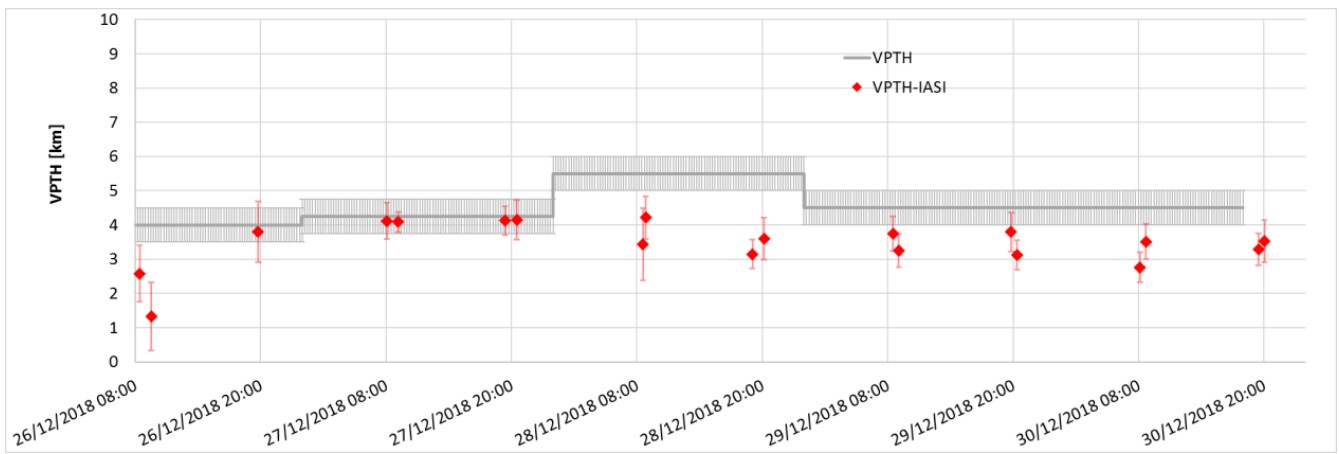

**Figure 8.** VPTH used for the SEVIRI, MODIS, VIIRS and TROPOMI retrievals (gray line) and VPTH used for IASI $SO_2$ retrievals.

As Figure 8 shows, while the agreement between the two can generally be considered to be very good, IASI estimates the mean VPTH to be lower than those obtained from the ground-based cameras, especially after 28 December. The differences between 1.3 and 2.5 km are in line with the expected uncertainties [83] but are those that induce the discrepancies between the $SO_2$ retrievals found.

### 4.2. $SO_2$ Flux

To avoid overly crowded plots that are difficult to interpret, the $SO_2$ flux cross-comparison is shown using two different figures: Figures 9 and 10 show SEVIRI/MODIS/VIIRS/TROPOMI and SEVIRI/TROPOMI/IASI/AIRS $SO_2$ fluxes, respectively. The SEVIRI curve is included in both plots for its continuity in time, while TROPOMI is present in both figures to show similarities and differences with multispectral and hyperspectral satellite sensor retrievals. All the fluxes are estimated using the traverse procedure and the same daily wind speeds (see Section 2). As Figure 9 shows, there is good agreement between the different curves, in particular for MODIS, VIIRS and TROPOMI. The main differences between LEO and GEO $SO_2$ fluxes lie in the region where the LEO $SO_2$ fluxes are not available because images are not present or where the plume dilution is significant, as in the distal part of the cloud (light red areas) and in single maxima and minima (see peak "P"). The lack of the single peak from SEVIRI may happen when the wind speed is higher than 3.3 m/s. As described in Section 3, when this condition is satisfied (in this case, about 6 m/s), a single short volcanic puff may not be detected. Another difference between fluxes is found on 27 December around 08:00 UTC (light green region), where the TROPOMI $SO_2$ flux is significantly lower than the fluxes obtained from the multispectral sensors. The reason for such a discrepancy could be the combination of the small plume width with high $SO_2$ columnar amounts in the region near the source (see Figure 5), and the wide TROPOMI spatial resolution compared with the multispectral instruments (see Figure 2).

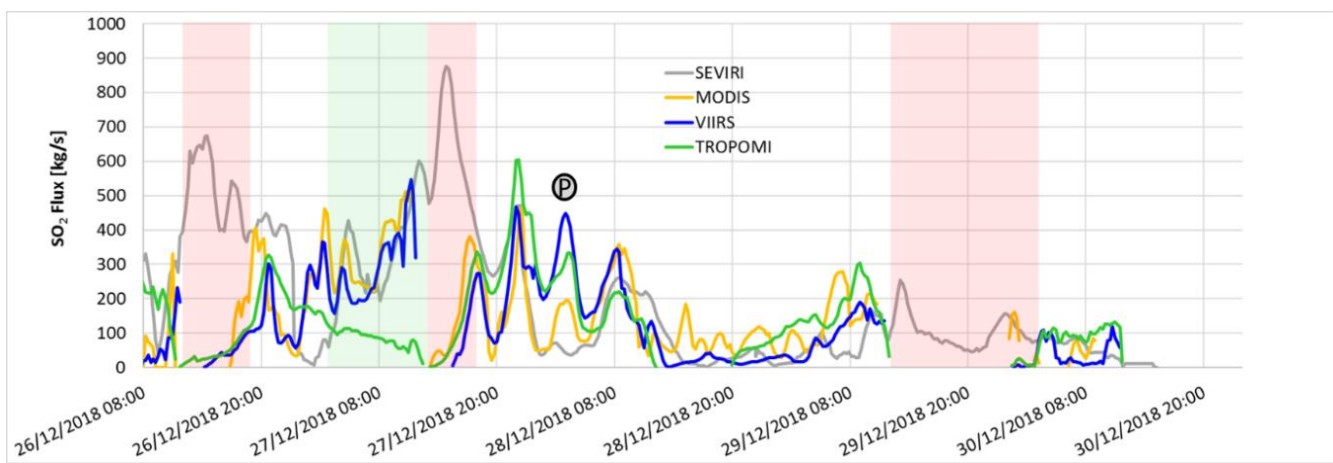

**Figure 9.** SO$_2$ fluxes obtained from SEVIRI, MODIS, VIIRS and TROPOMI measurements. The colored rectangle areas indicate the regions where the GEO and LEO retrieval discrepancies are significant. Light red areas: differences due to LEO images not available and/or where the plume dilution of the distal part of the cloud is significant. Light green area: differences due to the presence of volcanic ash. Point "P" is an example of an SO$_2$ peak not detected from SEVIRI.

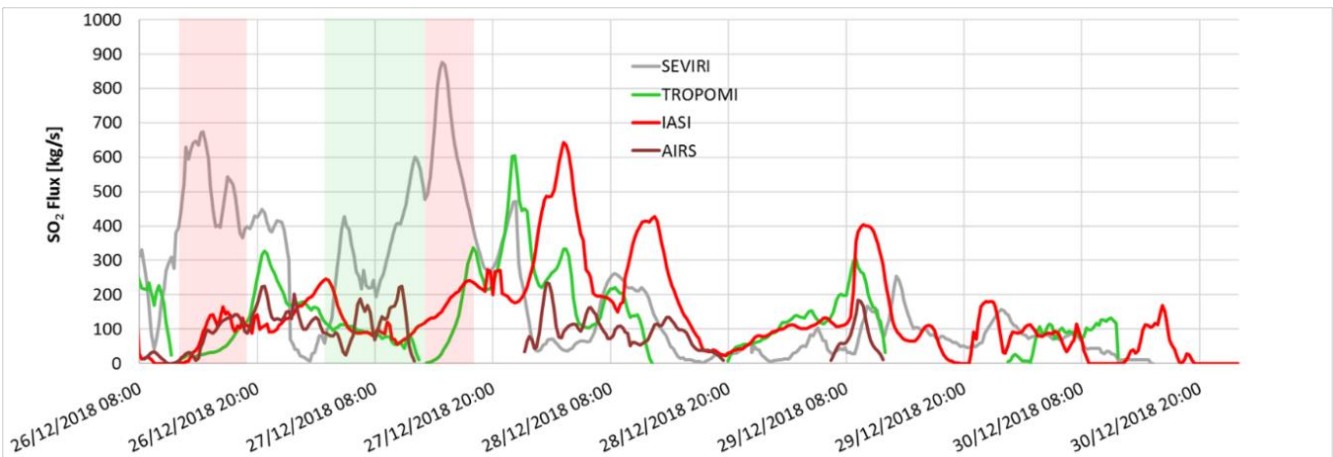

**Figure 10.** Cross-comparison between the SO$_2$ fluxes obtained from SEVIRI, TROPOMI, IASI and AIRS measurements. The colored rectangle areas indicate the regions where the GEO and LEO retrieval discrepancies are significant. Light red areas: differences due to LEO images not available and/or where the plume dilution of the distal part of the cloud is significant. Light green area: differences due to the presence of volcanic ash.

Figure 10 shows the SO$_2$ flux time series obtained from SEVIRI, TROPOMI, IASI and AIRS. As the figure emphasizes, and as expected from the analysis of the SO$_2$ total mass time series, the IASI flux is generally higher than the other. The figure indicates also that the low values of TROPOMI SO$_2$ flux, compared with multispectral instruments, found 27 December at around 08:00 UTC, is present also for IASI and AIRS. These latter sensors have ground pixels greater than TROPOMI.

## 5. SO$_2$ Flux Uncertainty Reduction and Assessment

Uncertainties in SO$_2$ flux are due to the uncertainty in SO$_2$ columnar abundance retrieval and wind speed (see Equation (1)). To reduce the SO$_2$ flux errors due to wind speed uncertainty, the 6 h wind speed profiles can be considered instead of the daily mean value. Figure 11 shows the comparison between the SEVIRI SO$_2$ fluxes obtained by considering the daily (gray dashed lines) and 6 h (gray solid lines) wind speeds, together with the daily (light blue dashed line) and 6 h (light blue solid lines) wind speed values used for the two computations. As the figure shows, and as expected, different wind speeds

lead to variation, in time and amount, of the $SO_2$ fluxes retrieved: the greater the wind speed difference, the greater the $SO_2$ flux time and amount discrepancy. However, being 30 km from the craters, the distance of the transect considered for the flux computation, a meaningful variation of the wind speed, doesn't significantly change the $SO_2$ emission time (note the $SO_2$ peaks that are all approximately in the same positions). However, a significant difference can be found in the $SO_2$ flux amount as shown, for example, between about 12:00 and 20:00 UTC on 27 December. Here, the high wind speed difference between the mean value (6.0 m/s) and the value in this time range (2.6 m/s) lead to significant differences in $SO_2$ amount.

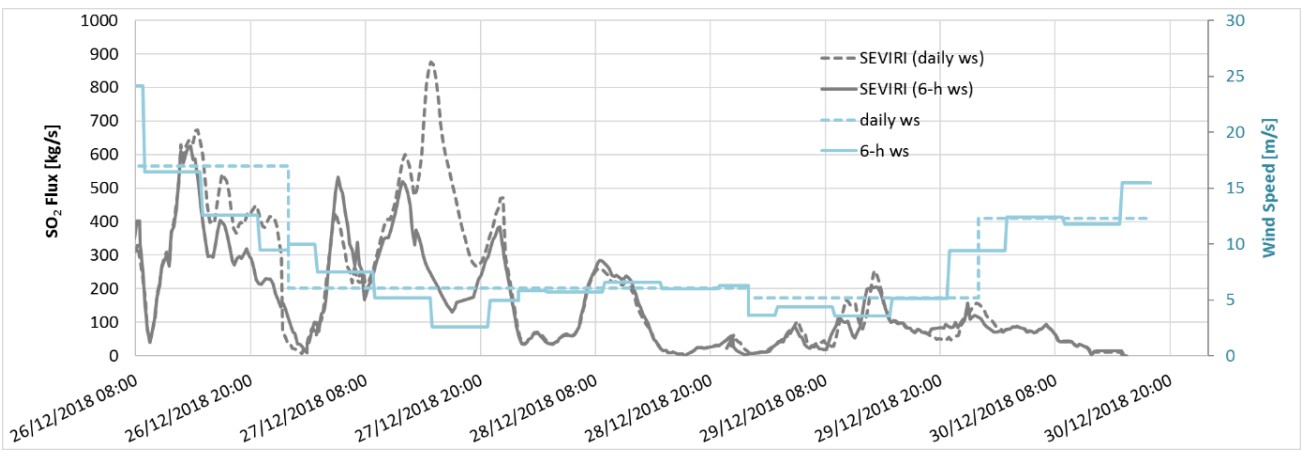

**Figure 11.** $SO_2$ flux obtained from SEVIRI 15-minute measurements, considering a fixed transect at 30 km from the vents. The gray dashed line indicates the flux obtained considering daily constant wind speed, while the gray solid line indicates the flux obtained considering the 6 h step wind speeds. The dashed and solid light blue lines indicate, respectively, the daily and 6 h wind speeds used for the $SO_2$ flux computations.

Because the wind speed is obtained directly from the VPTH, the uncertainty of the VPTH leads to uncertainty of the wind speed itself. Considering an uncertainty of $\pm 500$ m [81], the wind speeds are re-computed, and from those values, the $SO_2$ flux is derived (see Figure 12). As the figure shows, the VPTH uncertainty induced a wind speed variation of about $+/- 10\%$, which led to an $SO_2$ flux variation of about $+/- 20\%$.

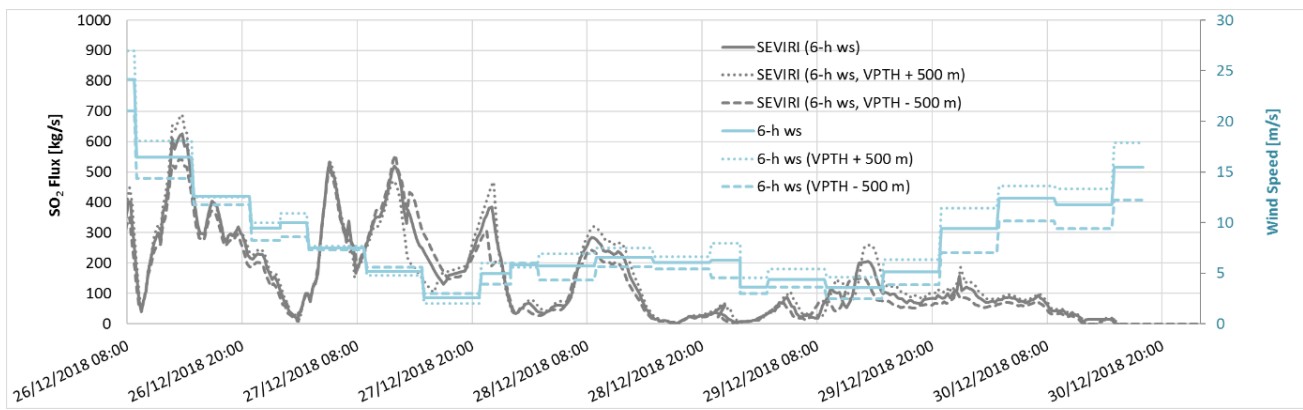

**Figure 12.** SEVIRI $SO_2$ fluxes computed considering different wind speeds obtained by considering an uncertainty of $+/- 500$ m of VPTH.

Finally, the last contribution to the $SO_2$ flux uncertainty can be attributed to the uncertainty of $SO_2$ columnar abundance retrieval. Figure 13 shows the total $SO_2$ flux

uncertainty due to the sum (in quadrature) of the 40% of errors of $SO_2$ columnar abundance and the 20% due to VPTH.

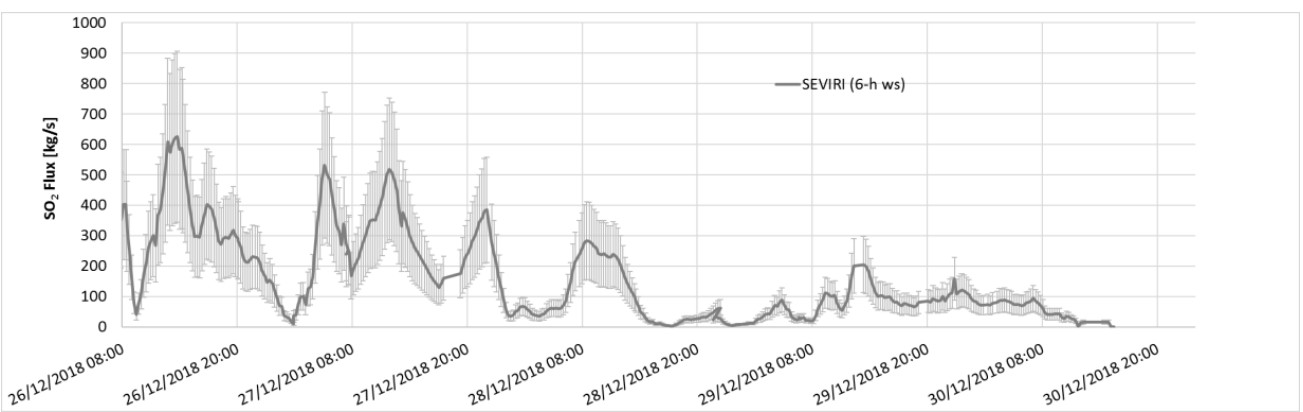

**Figure 13.** SEVIRI $SO_2$ flux. The uncertainty derives from the sum of the $SO_2$ columnar abundance and VPTH uncertainties.

## 6. Discussion

The results obtained confirm the higher sensitivity of the hyperspectral sensors TROPOMI and IASI with total volcanic $SO_2$ cloud areas, from two to four times the areas detected from the multispectral sensors SEVIRI, MODIS and VIIRS. The total masses obtained from the different instruments show a general good agreement with values within errors, except 28 December, for which the IASI total masses are significantly higher. In order to explain these differences, the volcanic plume top heights, used as input for the different $SO_2$ retrieval procedures, are compared. Even if the VPTH estimations can be considered in good agreement, the mean IASI estimations are generally lower than those obtained from the ground-based VIS cameras, in particular after 28 December. As demonstrated here, especially in the lower troposphere, differences of 1 or 2 km can have a large effect on the retrieved column. As an example, a plume at 4 km that is retrieved with IASI at a 3 km altitude underestimates the $SO_2$ column by a factor of ~2.7. Likewise, for a plume at 5 km retrieved at 4 km, the factor is around 2.3. These theoretical values are consistent with the differences observed in Figure 6.

Compared to the total mass, the $SO_2$ flux estimation is much more critical. If the $SO_2$ abundance is affected by the uncertainties in algorithm input parameters, the flux strongly depends also on the wind field, which is a function of both the height of the cloud and the distance from the crater. In this work, the $SO_2$ fluxes are computed by applying the "traverse" approach and considering a constant daily wind speed for all sensors. The SEVIRI flux is computed by fixing a transect perpendicular to the plume axis at 30 km from the crater and computing the flux every 15 min, while, for the LEO sensors, all the plume transects are considered. The main drawbacks of this strategy applied to geostationary data, are that some small volcanic puffs can be missed in the case of a wind speed greater than a characteristic value function of the SEVIRI spatial and temporal resolution. This problem can be greatly reduced by exploiting the SEVIRI rapid scan mode, by which the temporal frequency is increased from 15 to 5 min. Other reasons for the $SO_2$ flux discrepancies can be found in the presence of volcanic ash in the region near the vents and in the possible volcanic plume dilution in the distal part of the cloud. The ash presence could lead to an overestimation of the $SO_2$ retrievals by the multispectral systems, and vice versa for the UV sensors. The dilution of the cloud in the distal part of the volcanic cloud, on the other hand, leads to an underestimation of the $SO_2$ flux obtained. Finally, the variation of the wind direction could also lead to uncertainties in the LEO $SO_2$ flux computations rather than GEO estimations. However, the characteristics of the wind field during the days of the measurements in the relatively small area considered guarantee that this aspect is negligible.

Despite all the described criticalities, the results for $SO_2$ flux show a good agreement between all the instruments, with peaks and valleys well reproduced. As expected, the IASI flux of 28 December is higher than that obtained from the other sensors. Moreover, the significant difference found the morning of 27 December between multispectral (high values) and the hyperspectral (low values) can be attributed to the combination between the small plume width with high $SO_2$ columnar amounts in the region near the source and the wide hyperspectral spatial resolution. Finally, the results also confirm the ability of the LEO sensors to provide information on $SO_2$ flux during large time intervals and, for daily sensors such as TROPOMI, to also give information on $SO_2$ flux during the night.

A complete error assessment of $SO_2$ flux retrievals using SEVIRI data is also realized by considering uncertainties of $SO_2$ abundance and wind speed. At first, instead of a single average daily wind speed value, four values, deriving from the ARPA profiles of 00, 06, 12 and 18 UTC, were considered. Different wind speeds lead to variation, in time and amount, of the $SO_2$ fluxes retrieved. This difference doesn't induce significant changes in the $SO_2$ emission time, with the transect distance from the craters being only 30 km, but leads to a significant difference in $SO_2$ flux amount. The more than double wind speed difference between about 12 and 20 UTC on 27 December leads to a significant difference in the $SO_2$ obtained. Moreover, the influence of volcanic cloud height uncertainty on wind speed computation is studied. Taking into account a VPTH uncertainty of $+/-$ 500 m, the wind speeds are recomputed and then the $SO_2$ fluxes. The VPTH uncertainties considered lead to a mean wind speed variation of about $+/-$ 10%, which corresponds to a mean $SO_2$ flux variation of about $+/-$ 20%. Finally, considering the linear relationship between mass and flux, the $SO_2$ columnar amount uncertainty is reported directly for $SO_2$ flux estimation. The total contribution due to $SO_2$ columnar abundance and VPTH leads to a total $SO_2$ flux uncertainty of about 45%.

## 7. Conclusions

In this paper, the $SO_2$ masses and fluxes obtained from several GEO (SEVIRI) and LEO (MODIS, VIIRS, TROPOMI, IASI, AIRS) satellite sensors are compared. As a test case, the Christmas 2018 Etna eruption, from 26 to 30 December, is considered. In this time range, the eruption was tropospheric, with high $SO_2$ and low ash contents. The large amount of data available, together with the different instrument characteristics and $SO_2$ retrieval strategies, make this cross-comparison particularly significant.

The results confirm the higher sensitivity of TROPOMI and IASI with volcanic cloud detection in distal areas where AIRS and multispectral instruments are not able to see.

The comparison between the total $SO_2$ masses shows a general good agreement with values within errors, except for 28 December, for which the IASI total masses are significantly higher. This discrepancy is due mainly to the difference between the VPTHs used by IASI and by all the other satellite instruments. In the troposphere, differences between 1.3 and 2.5 km, even if in line with expected uncertainties, are those that induce the discrepancies found in $SO_2$ total mass estimations.

The $SO_2$ fluxes, obtained considering the "traverse" approach, show a good agreement, except in the regions where the dilution is significant (distal part of the volcanic cloud), where the plume has high $SO_2$ amounts and ash influence is significant (near the source) and where the differences in VPTH induce differences in $SO_2$ content. The results confirm also the ability of the LEO sensors to provide information on $SO_2$ flux during large time intervals and, for daily sensors such as TROPOMI, to give information during the night.

Finally, the complete error assessment of $SO_2$ flux retrievals using SEVIRI data is realized. The total $SO_2$ flux uncertainty—the sum of the uncertainties of $SO_2$ abundance and wind speed—is estimated to be about 45%.

**Author Contributions:** Conceptualization and work coordination, S.C.; methodology, S.C., L.G.; SEVIRI and MODIS data processing, S.C., L.G., D.S., L.M., F.P.; VIIRS data processing, V.J.R.; TROPOMI data processing, N.T., H.B.; IASI data processing, L.C.; AIRS data processing, A.J.P.; data

analysis and cross-comparison, all authors; writing—original draft preparation, all authors. All authors have read and agreed to the published version of the manuscript.

**Funding:** This research was supported by the ESA project VISTA (Volcanic monItoring using Sen-Tinel sensors by an integrated Approach), grant number 4000128399/19/I-DT, by the INGV project Pianeta Dinamico (CUP D53J19000170001) supported from MIUR ("Fondo finalizzato al rilancio degli investimenti delle amministrazioni centrali dello Stato e allo sviluppo del Paese", legge 145/2018) -Task V3–2021 and from INGV Department strategic project 2019 IMPACT (a multidisciplinary In-sight on the kinematics and dynamics of Magmatic Processes at Mt. Etna Aimed at identifying preCursor phenomena and developing early warning sysTems). L. Clarisse, a research associate supported by the Belgian F.R.S-FNRS. BIRA-IASB, acknowledges financial support from ESA S5P MPC (4000117151/16/I-LG), Belgium Prodex TRACE-S5P (PEA 4000105598) projects. This paper contains modified Copernicus data (2018) processed by BIRA-IASB. The V.J. Realmuto research was conducted at the Jet Propulsion Laboratory, California Institute of Technology, under contract to the National Aeronautics and Space Administration (NASA).

**Institutional Review Board Statement:** Not applicable.

**Informed Consent Statement:** Not applicable.

**Data Availability Statement:** The data presented in this study are available on request from the corresponding author.

**Acknowledgments:** The authors would like to thank EUMETSAT, ESA and NASA for providing satellite data.

**Conflicts of Interest:** The authors declare no conflict of interest.

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
