# Peer review of "Tropospheric Volcanic SO2 Mass and Flux Retrievals from Satellite. The Etna December 2018 Eruption"

_remotesensing, doi:10.3390/rs13112225_

Round 1

Reviewer 1 Report

Review of “Tropospheric volcanic SO2 mass and flux retrievals from satellite. The Etna December 2018 eruption”

The manuscript compares data from six satellites considering SO2 area, mass, and flux. Further uncertainty analysis is given for SEVIRI considering wind speed and plume top height. The subject of the work is an eruption period for several days at Etna in Italy. The writing is generally clear, including good discussion/conclusions. It is important to figure out how to best use these satellite products to quantify SO2 from volcanoes and associated hazards. I have only a few comments:

An immediate thing noticed while reading is that the figures are blurry. It should be checked whether they were inserted with at least 300 dpi or if perhaps there was a PDF conversion issue.

Also note that Figures 2 & 6 are each shown twice

On Fig 9 and 10, were these plotted using smoothed lines? Usually straight lines are used to indicate there is no information in-between. It is not obvious to me even where the overpasses are since it looks like there is data throughout including for TROPOMI, etc.

Fig 10 - At first look, SEVIRI is almost uncorrelated with the others. The red styling for plume dilution should be added as in Fig 9

While six satellites are already investigated, any comment on why OMI and OMPS are excluded?

L255 ‘When acquisitions of the same satellite sensor appear in rapid succession (see Figure 3, for MODIS, VIIRS, TROPOMI and IASI), the flux is computed as the mean of the fluxes obtained from the single images at the same time.’ – Related to my question about Fig9-10, I am note sure if this sentence is accurate. Many more data points are indicated in the flux plots than correspond to times on Figure 3

L300 aboundance => abundance

L303 abilty => ability

L303 significant different => significantly different

L421 contribute => contribution

L519 remove ‘please add’

Reviewer 2 Report

Authors presented material on tropospheric volcanic SO2 mass and flux retrievals using various satellite data for Etna eruption event in 2018. Material is interesting. A lot of data was collected. Authors clearly presented data, methods and analysis discussing errors and limitations. From technical point of view material is consistent, however edition of it is really poor, please work it out.

For instance:

Fig. 1 I saw it somewhere, authors used this figure in another article, please modify.

Fig. 2  looks to be double.

Fig.4 is not clear, please clarify and provide unified section; the same for fig. 3 which should be placed after fig.4.

Fig. 6 looks to be double, please correct.

Something happened to page numbers, please order it.

Reviewer 3 Report

Dear authors, the paper is interesting, but it is more written as scientific popular paper, then real scientific paper. There is lack of technical details and also is not well described scientific approach. It is necessary to describe better, how ground/terrestrial data about content of SO2 was measure, methods, how this was compared with data obtained from satellite (mainly with focus on spatial distribution). This scientific and technological approach is missing in paper and without this, the results cannot be interpreted as real scientific results.

Round 2

Reviewer 3 Report

Dear authors, the study only compare different satellites,  but there is no comparison with reality and no ground trough. This I see as important fact. Without this seems results without real verification.